# Divergent evolution of male-determining loci on proto-Y chromosomes of the housefly

Xuan Li [1,2] ✉, Sander Visser [1], Jae Hak Son[3], Elzemiek Geuverink[1], Ece Naz Kıvanç [4], Yanli Wu[1,5], Stephan Schmeing[4,6], Martin Pippel [7], Seyed Yahya Anvar[8], Martijn A. Schenkel [1,9], František Marec [10], Mark D. Robinson [4,6], Richard P. Meisel [11], Ernst A. Wimmer [5], Louis van de Zande[1], Daniel Bopp [4] & Leo W. Beukeboom [1]

Houseflies provide a good experimental model to study the initial evolutionary stages of a primary sex-determining locus because they possess different recently evolved proto-Y chromosomes that contain male-determining loci (*M*) with the same male-determining gene, *Mdmd*. We investigate *M*-loci genomically and cytogenetically revealing distinct molecular architectures among *M*-loci. *M* on chromosome V ($M^V$) has two intact *Mdmd* copies in a palindrome. *M* on chromosome III ($M^{III}$) has tandem duplications containing 88 *Mdmd* copies (only one intact) and various repeats, including repeats that are XY-prevalent. *M* on chromosome II ($M^{II}$) and the Y ($M^Y$) share $M^{III}$-like architecture, but with fewer repeats. $M^Y$ additionally shares $M^V$-specific sequence arrangements. Based on these data and karyograms using two probes, one derives from $M^{III}$ and one *Mdmd*-specific, we infer evolutionary histories of polymorphic *M*-loci, which have arisen from unique translocations of *Mdmd*, embedded in larger DNA fragments, and diverged independently into regions of varying complexity.

Sex determination mechanisms are highly diverse and undergo rapid turnover in evolution. In insects, sex is determined by a hierarchical cascade in which upstream genes regulate the activity of downstream genes. New sex determination genes can be added sequentially or emerge to replace old sex-determining genes at the top of the cascade[1]. Several primary signal genes have been characterized in insects (reviewed in ref. [2]). These genes share remarkably little homology, suggesting that they have arisen independently. As of yet, we know very little about how novel sex-determining genes evolve,

both in terms of neofunctionalization of existing sequences and the associated genomic rearrangements.

The emergence of a novel sex determination gene will affect its genomic surroundings. A dominant male- or female-determining gene will always be hemizygous. A specific prediction of the canonical sex chromosome evolution model is that a sex-determining region will undergo progressive recombination suppression[3–7]. Suppressed recombination is predicted to prevent gene flow between proto-sex chromosomes so that the sex-determining region can be sex-limited

[1]Groningen Institute for Evolutionary Life Sciences, University of Groningen, Groningen, The Netherlands. [2]Department of Organismal Biology – Systematic Biology, Evolutionary Biology Centre, Uppsala University, Uppsala, Sweden. [3]Department of Genetics, Rutgers, The State University of New Jersey, Piscataway, NJ, USA. [4]Department of Molecular Life Sciences, University of Zürich, Zürich, Switzerland. [5]Department of Developmental Biology, Johann-Friedrich-Blumenbach Institute of Zoology and Anthropology, Göttingen Center of Molecular Biosciences, University of Göttingen, Göttingen, Germany. [6]SIB Swiss Institute of Bioinformatics, University of Zurich, Zürich, Switzerland. [7]Department of Cell and Molecular Biology, National Bioinformatics Infrastructure Sweden (NBIS), Science for Life Laboratory, Uppsala University, Uppsala, Sweden. [8]Department of Human Genetics, Leiden University Medical Center, Leiden, The Netherlands. [9]Department of Biology, Georgetown University, Washington, DC, USA. [10]Institute of Entomology, Biology Centre of the Czech Academy of Sciences, České Budějovice, Czech Republic. [11]Department of Biology and Biochemistry, University of Houston, Houston, TX, USA. ✉e-mail: lx1290@hotmail.com

and thus effectively hemizygous. This leads to mutation accumulation, transposon insertion, and other structural rearrangements that increase the sequence divergence between the sex chromosome pair[6]. Validation of this model requires more detailed knowledge of the genomic organization of sex determination loci as well as their neighboring regions.

The housefly (*Musca domestica*) has a polymorphic sex determination system[8,9] that has been instrumental for investigating early processes of sex chromosome evolution[10–12]. A male development trajectory can be induced by a dominant male-determining locus *M* on the Y chromosome[8,13]. However, an *M*-locus can also be present on any of the five autosomes or on the X chromosome[14–20]. All chromosomes carrying an *M*-locus appear to be of recent origin[21], suggesting that they are "proto-Y" chromosomes. *M* is needed to break the autoregulatory splicing loop of the female-promoting *transformer* (*Mdtra*) gene to allow for male development. We previously identified *Musca domestica male determiner* (*Mdmd*), which is a paralogue of the generic splice factor gene *nucampholin* (*Mdncm*), as a male-determining gene of the housefly[13]. *Mdmd* is present in *M*-loci on chromosomes II ($M^{II}$), III ($M^{III}$), and V ($M^V$) and the Y chromosome ($M^Y$). However, the structures of the various *M*-loci are both diverse and complex[13], providing a unique opportunity to investigate the primary evolution of sex-determining regions and sex chromosomes.

Here, we show the genomic organization of *Mdmd*-containing *M*-loci on various proto-Y chromosomes in the housefly. We find different levels of complexity for these loci, reflected in the number of *Mdmd* copies and intervening sequences. $M^V$ contains only two expressed *Mdmd* copies in palindromic structure. In contrast, $M^{III}$ contains numerous *Mdmd* copies of which only one is functional, and some intervening sequences that represent non-male-specific repeats. $M^{II}$ and $M^Y$ share $M^{III}$-like architecture albeit with fewer repeats. Together, our genomic and cytogenetic results point to a common origin but distinctive evolution of *M*-loci.

## Results

In the following text, genomic regions with a dominant male-determining locus are referred to as *M*-loci with a Roman numeral superscript indicating on which chromosome the locus is found, i.e., $M^{III}$ is the *M*-locus on chromosome III. Non-italic letter M with an Arabic number is used to describe housefly strains or genomic datasets (e.g., M5 is a strain with $M^V$ and females without *M*). *Mdmd* is the male-determining gene within all of the *M*-loci investigated.

### Complexity and chromosomal location of *M*-loci

Previous comparison of $M^{II}$, $M^{III}$, and $M^Y$ revealed that they all contain at least one complete *Mdmd* gene and various incomplete copies[13]. In order to estimate structural divergence between *M*-loci, we performed Illumina sequencing on strains M3 (males that carry $M^{III}$ and females without an *M*), M5 (males that carry $M^V$), and M2 (males that carry $M^{II}$). We also used published Illumina reads of three $M^Y$ strains of different geographical origin[21], namely *aabys* (laboratory generated strain with $M^Y$), A3 (strain with $M^Y$ that was derived from a collection in Marshall County, Alabama, USA in 1998), and LPR (strain with $M^Y$ that was originally collected near Horseheads, New York, USA). See Table 1 in Methods for an overview of the strains used and type of genomic data analyzed in this study. We determined the read mapping coverage per base pair of *Mdmd* relative to that of three single-copy reference genes: *Mdtra*, *yellow* (*MdY*), and *asense* (*Mdase*), based on the Illumina sequence data. Such coverages essentially represent *Mdmd* copy numbers in the tested *M*-loci and, therefore, are indicative of differences in the sizes of *M* genomic loci. The two M3 male datasets had the highest average coverage (~41.44 and ~41.88) indicating the highest copy number of *Mdmd* in $M^{III}$, whereas these were lowest in the M5 male dataset (~2.38, Fig. 1). Coverages in the M2 male dataset (~18.58) and two MY datasets (*aabys*-male, ~19.62; A3-male, ~19.74) were

approximately half of the $M^{III}$ value. Interestingly, one MY dataset, LPR-male, had higher average coverage (~34.78) than the other two MY datasets, and almost as large as the $M^{III}$ coverage. Taken together, these data reveal that the number of *Mdmd* sequences vary considerably both between and within *M*-containing chromosomes.

To identify the cytogenetic localization of *M*-loci on the male-determining chromosomes of various housefly strains, we performed fluorescence in situ hybridization (FISH) with an *Mdmd*-specific probe and karyogram obtained from the brain tissues of third-instar larvae. *M*-loci on chromosome II and III as well as on the X and Y chromosomes were successfully localized by detecting a single signal, indicating the presence of clustered *Mdmd* sequences on these chromosomes (Fig. 2a, b; Supplementary Fig. 1). The $M^{II}$, $M^{III}$, and $M^Y$-loci were all located in the pericentromeric regions on the short arm of the chromosomes. *M* on the X ($M^X$) was located on one arm of the chromosome but was not pericentromeric. Using samples from multiple laboratory strains, as well as wildtype strains from Spain, Italy, and the Netherlands, the *M*-loci were localized at the same position on their respective chromosomes, regardless of strain origin (Supplementary Fig. 1), suggesting a single evolutionary origin of each of these *M*-loci. In the M5 samples, we did not detect a hybridization signal for $M^V$ (Fig. 2c) although PCR assays were positive for the presence of *Mdmd*. This is likely due to the low resolving power of the *Mdmd*-specific probe, which is insufficient to generate a detectable signal if few *Mdmd* copies are present.

As the results indicated that the genomic sizes of $M^{III}$ and $M^V$ were the most distinct, we proceeded with these two *M*s. The housefly reference genome was generated from female genomic DNA[22], and we therefore assembled male genomes from Pacbio SMRT sequencing of the strains M3 (~116× total coverage) and M5 (~161× total coverage) in order to obtain genomic sequences of $M^{III}$ and $M^V$. Both of the assembled genomes were ~1.3 Gb in size; the M3 genome assembly consists of 11,176 contigs with an N50 of ~617.5 kb, and the M5 genome assembly contains 4327 contigs and has an N50 of ~7800.3 kb. The haploid housefly genome is estimated to be ~1 Gb[23], suggesting that our assemblies either contain unresolved allelic variation or phased assembly of the proto-X and proto-Y chromosomes. According to BUSCO analysis, both genomes have ~99% complete matches to 3285 universal single-copy orthologs in dipteran lineages. In addition, we investigated an $M^Y$ of the *aabys* strain, in order to compare autosomal *M*-loci to *M* from a morphologically differentiated XY pair. We obtained $M^Y$ sequences by generating an assembly (*aabys*-male) with Pacbio sequencing data, which was polished with Illumina sequencing data of males from the same strain (~13× coverage). Details of the three assemblies can be found in Supplementary Table 1.

### Genomic structure of the $M^V$

We first screened the M5 genome for *Mdmd*-containing contigs. We identified one ~4 Mb contig (tig00004758; Fig. 3a, Supplementary Table 2, referred to as $M^V$-contig) that contained two intact copies of *Mdmd* in opposing orientation, approximately 4.7 kb apart (Fig. 3a). This is in line with the estimated ~2× coverage of *Mdmd* for the M5 genome. Only a single synonymous nucleotide substitution, located in exon 2, was found between these two *Mdmd* copies. Based on this nucleotide difference, we identified transcripts of both *Mdmd* copies (Supplementary Fig. 2), demonstrating that both are expressed. The $M^V$-contig is the only contig of the M5 genome that harbors *Mdmd* sequences, which indicates that $M^V$ has a compact architecture.

To determine the borders of $M^V$, we examined whether parts of the $M^V$-contig were covered by sequences derived from the non-*M*-containing chromosome V of the M5 and M3 genomes. We identified one such contig in the M5 genome (tig00002184, referred to as non-$M^V$-contig$^{M5}$) and one in the M3 genome (contig7533, referred to as non-$M^V$-contig$^{M3}$). Alignment of the $M^V$-contig and both non-$M^V$-contigs revealed the sequences shared between chromosome V with and

**Table 1 | Overview of the strains and genomic datasets used in the current study**

| Strain | M-locus[a] | Origin | Usage | Genomic dataset | Accession No. | Reference |
|---|---|---|---|---|---|---|
| M2 | $M^I$ | Laboratory strains | FISH localization | M2-male (Illumina reads, ~10.5 Gb) | SRX21801162 | Current study |
| M3 | $M^{III}$ | | Genomic analysis | Genome M3 (Assembled genome, ~1.3 Gb); M3-male_1 (Illumina reads, ~46.2 Gb); M3-male_2 (Illumina reads, ~50.0 Gb); M3-female_1 (Illumina reads, ~64.6 Gb); M3-female_2 (Illumina reads, ~102.8 Gb); | JAVQME000000000 SRX21801164 SRX21801165 SRX21801166 SRX21801167 | |
| M5 | $M^{III}$ | | | Genome M5 (Assembled genome, ~1.3 Gb) M5-male (Illumina reads, ~12.4 Gb) | JAVVNY000000000 SRX21801163 | |
| aabys | $M^Y$ | | | Genome aabys-male (Assembled genome, ~893.7 Mb) | JAZGUT000000000 | |
| | | | | Genome aabys (Assembled genome, ~750.4 Mb) aabys-female (Illumina reads, ~14.7 Gb) aabys-male (Illumina reads, ~16.7 Gb) | GCA_000371365.1 SRX2154714 SRX2154715 | refs. 20,21 |
| A3 | $M^Y$ | Marshall County, Alabama, USA | Genomic analysis | A3-female (Illumina reads, ~15.4 Gb) A3-male (Illumina reads, ~13.5 Gb) | SRX2154716 SRX2154717 | |
| LPR | $M^Y$ | Horseheads, New York, USA | | LPR-female (Illumina reads, ~12.4 Gb) LPR-male (Illumina reads, ~10.4 Gb) | SRX2154718 SRX2154719 | |
| ITA1 | $M^{III}$, $M^Y$ | Altavilla Silentina, Italy | FISH localization | N/A | N/A | Sander Visser & Leo W. Beukeboom (unpublished) |
| ITA3 | $M^I$, $M^{III}$, $M^X$ | Castelianeta marina, Italy | | | | Current study |
| SPA1 | $M^X$ | Catalonia, Spain | | | | ref. 19. |
| SPA2 | $M^{II}$, $M^X$ | | | | | |
| SPA3 | $M^{III}$ | | | | | |
| SPA4 | $M^I$, $M^{II}$, $M^X$ | | | | | |
| SPA5 | $M^{III}$ | | | | | |
| NL1 | $M^Y$ | Gerkesklooster, the Netherlands | | | | |

[a]Male-determining locus are referred to as italic M with a Roman numeral superscript indicating on which chromosome the locus is found, i.e., $M^{III}$ is the M-locus on chromosome III.

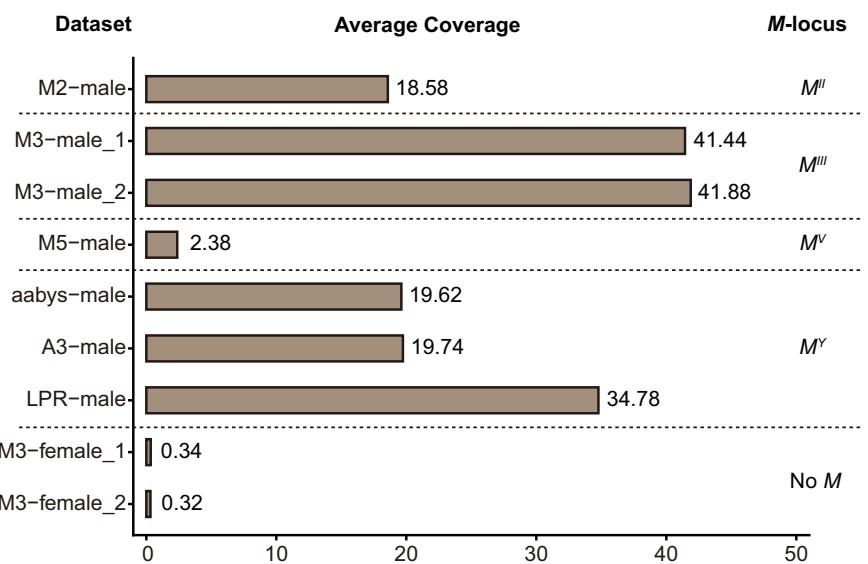

**Fig. 1 | The average coverages of *Mdmd* gene in different datasets.** Coverage rates in female genomes are included to account for off-target mapping to the paralogous gene *Mdncm* and the calculated average coverages in two M3 female Illumina datasets turned out to be negligible. Average coverages demonstrate that the number of *Mdmd* sequences are highest in $M^{III}$, intermediate in $M^Y$ and $M^{II}$, and lowest in $M^V$. Source data are provided as a Source Data file.

without the *M*-locus (Fig. 3b). The ~31 kb sequence, which exists only on the $M^V$-contig and includes the two opposing *Mdmd* copies, can thus be considered as the complete $M^V$ locus. $M^V$ is integrated in a tandem repeat block with a repeat unit ~10 kb shared between the $M^V$-contig and both non-$M^V$-contigs.

$M^V$ has a palindromic structure (Fig. 3c, d) with the two arms separated by a 3046 bp spacer sequence. Part of the spacer sequence shows homology to *reverse transcriptase* in *Lasius niger* (Accession: KMQ86458) and *Drosophila simulans* (Accession: AAS13459), and partially overlaps with a predicted housefly non-coding RNA (Accession: LOC109613599). At each end of the spacer, mariner-like terminal repeats are present and extend into the palindrome arms. Although some small variations and a few deletions/insertions were found, high sequence identity was observed between the palindromic arms. Based on the distribution of single-copy BUSCOs, similar synteny was observed between the $M^V$-contig and *Drosophila melanogaster* chromosome 2 R (Muller element C), which corresponds to chromosome V in the housefly[21], confirming the chromosomal location of $M^V$ (Fig. 3e).

RepeatModeler recognized large blocks of tandem repeats (Fig. 3c, d) located palindromically at the distal parts of $M^V$ as interspersed repeats, reminiscent of transposable elements. Moreover, at the ends of the $M^V$ locus, we identified Terminal Inverted Repeats (TIRs) and a 9-bp long direct repeat (TTTTAGGTT), which flanks the TIRs and is present as a single copy in the non-$M^V$-contigs (Fig. 3f). This direct repeat sequence thus resembles a target site duplication of a transposition event. Interestingly, by examining 16 independent genomic regions containing a similar stretch of interspersed repeats and palindromic structures, we could identify almost identical TIRs to $M^V$ and respective target site duplications (Fig. 3g; Supplementary Fig. 3).

**The complex structure of $M^{III}$**

The architecture of $M^{III}$ is distinctive from $M^V$, $M^{III}$ contains only a single functional *Mdmd* gene, and also a high number of additional truncated copies of *Mdmd*. We identified two contigs in the M3 genome carrying *Mdmd* sequences (Contig6762, ~202 kb, referred to as $M^{III}$-contig-1; Contig7871, ~389 kb, referred to as $M^{III}$-contig-2; Fig. 4a, Supplementary Table 2). The *Mdmd* sequences scattered across both contigs indicate the large size of $M^{III}$. We could not

determine the exact borders of $M^{III}$ because we did not find corresponding sequences derived from the non-*M*-containing chromosome III when performing a BLAST search with the terminal sequences of both $M^{III}$-contigs against the M3 and M5 genomes. Thus, the $M^{III}$ locus might extend beyond the length of the two $M^{III}$-contigs. As these two $M^{III}$-contigs share high sequence similarity at one end of each contig (Fig. 4c), they are presumably connected via the overlap. We, therefore, consider both $M^{III}$ contigs as part of one continuous locus encompassing in total ~591 kb, which is more than twenty times larger than $M^V$.

Unlike $M^V$, the $M^{III}$-contigs do not have a palindromic structure, but instead, contain highly replicated sequences that mostly occur in a tandem (head-to-tail) fashion and largely cluster together. Even though the majority of the repetitive sequences are truncated *Mdmd* copies (approximately 13% of $M^{III}$-contig-1 and 26% of $M^{III}$-contig-2), non-*Mdmd*-associated repeats were also identified (Fig. 4b, gray boxes). The *Mdmd* copies and the additional repeats do not show any obvious replication pattern, as the repeated sequences vary in length as well as start and end points (Supplementary Fig. S4).

In $M^{III}$, we identified 88 *Mdmd* copies, of which only one represents an intact open reading frame (ORF) (Supplementary Data 1, No. 36). To identify genes in $M^{III}$ other than *Mdmd*, we used sequences of $M^{III}$-contigs as queries in a BLAST search against the NCBI *M. domestica* (Taxid: 7370) Nucleotide Collection database. We found many matches to uncharacterized mRNA and ncRNA sequences as well as 17 matches to predicted genes (Supplementary Table 3). For each of these partially matched genes, we could identify $M^{III}$-independent contigs with higher sequence similarity, which indicates that the non-*Mdmd* genes in $M^{III}$ are likely degenerated pseudo-copies of genes present elsewhere in the genome. None of these genes have been reported to be involved in sex-determination. Using RepeatModeler, we identified 136 instances of known transposable elements in $M^{III}$-contig-1 and 196 in $M^{III}$-contig-2 (Supplementary Table 4). In nine cases, the transposable element resides within *Mdmd* copies (Supplementary Data 2), which indicates that some transposons accumulated after *Mdmd* replication in $M^{III}$.

**$M^Y$ shows homology to both $M^{III}$ and $M^V$**

In the *aabys*-male genome, we retrieved 4 contigs, $M^Y$-contig-1 (contig_6317_pilon), $M^Y$-contig-2 (contig_2268_pilon), $M^Y$-contig-3

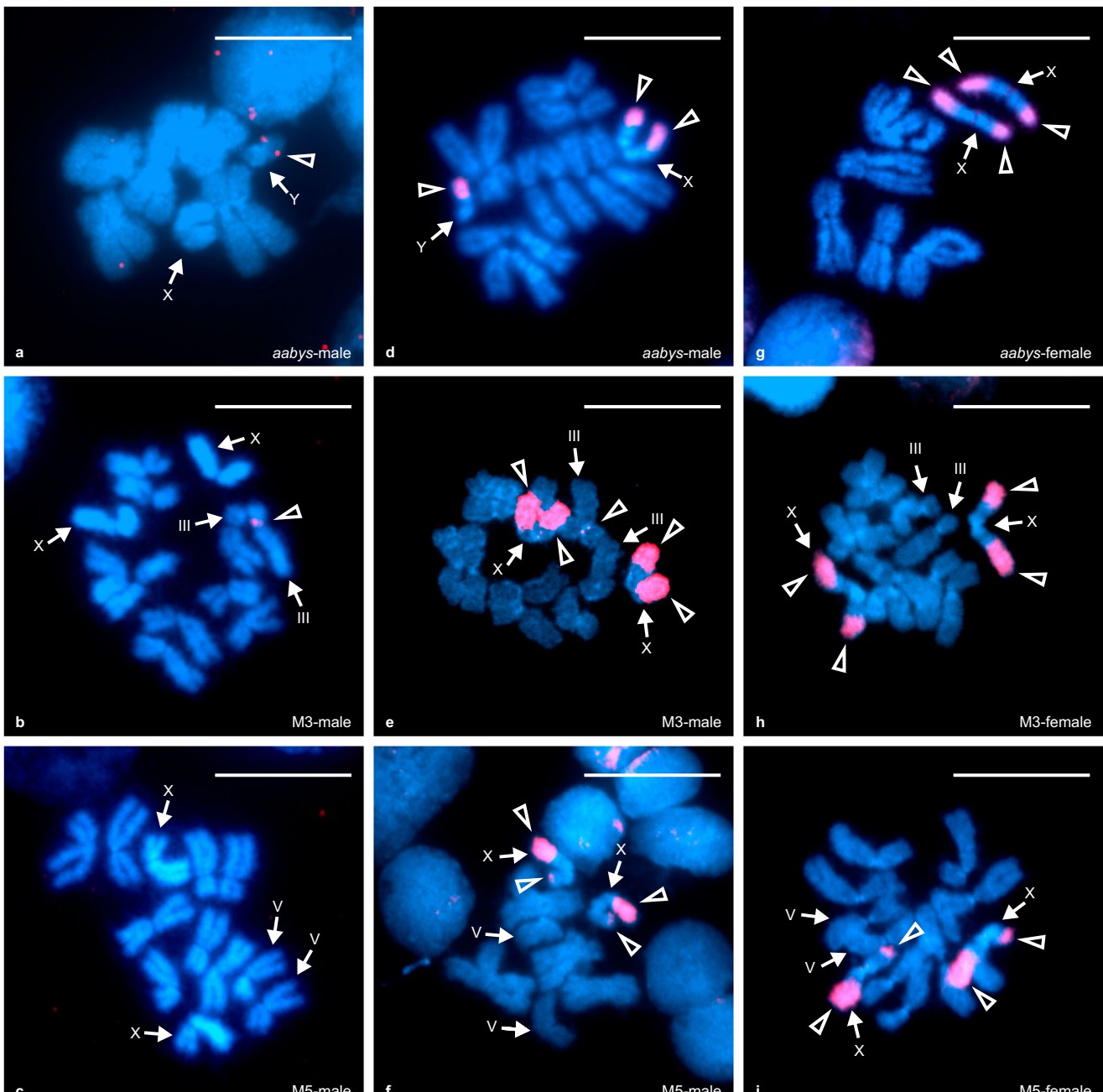

**Fig. 2 | FISH localization of *M*-loci and sex chromosome-associated repeat regions. a**, **b** Using an *Mdmd*-specific probe, $M^Y$ and $M^{III}$ were localized to peri-centromeric regions of the Y chromosome and chromosome III respectively. **c** $M^V$ was not detected by the *Mdmd* probe due to insufficient gene copy numbers. **d**–**i** Using a probe containing a mixture of amplified $M^{III}$ sequences including *Mdmd* and non-*Mdmd* intervening sequences, the *M*-locus and the *M* and sex chromosome-located (MAS) regions of the XY chromosome pair were localized. The signals of the mixed probe on the Y chromosome cover most parts of the short arm and merge with the $M^Y$ signal. The signal of the mixed probe on the X chromosome mark at the ends of both arms. Positive signals are shown in red and indicated by open triangles, chromosomes are indicated by arrows. Metaphase chromosomes are shown in blue. Signals were only considered as a successful hybridization if they were observed with consistent chromosomal locations on at least 20 metaphase nuclei on each slide. For each strain, 2–3 individuals were tested to ensure reproducibility. Scale bar: 10 µm.

(contig_2269_pilon), and $M^Y$-contig-4 (contig_12930_pilon), that contain *Mdmd* sequences, which were considered as $M^Y$ sequences (Fig. 4d, Supplementary Table 2). Note that because of the low coverage of our Pacbio dataset, we likely did not capture all sequences of $M^Y$. The $M^Y$-contigs are informative as one of them ($M^Y$-contig-4) appears to contain an intact copy of *Mdmd* although with several indels that may be due to the low quality of the assembly. Upon examining all four $M^Y$-contigs, many truncated *Mdmd* copies are observed in a tandem fashion similar to $M^{III}$

(Fig. 4d). Further homology is found for $M^Y$-contigs to various parts of $M^{III}$-contigs (Fig. 4e, f, g). $M^Y$-contig-1 and $M^Y$-contig-2 align with two separate regions on $M^{III}$-contig-1 which are ~50 kb apart (Fig. 4g). $M^Y$-contig-3 and $M^Y$-contig-4 align to a continuous region on $M^{III}$-contig-2, which cover both upstream and downstream sequences of the intact *Mdmd* gene. Thus, $M^Y$ shares a similar sequence architecture with $M^{III}$, which is also demonstrated via independent alignment of Illumina *aabys* male reads to $M^{III}$ (see below, Fig. 4i).

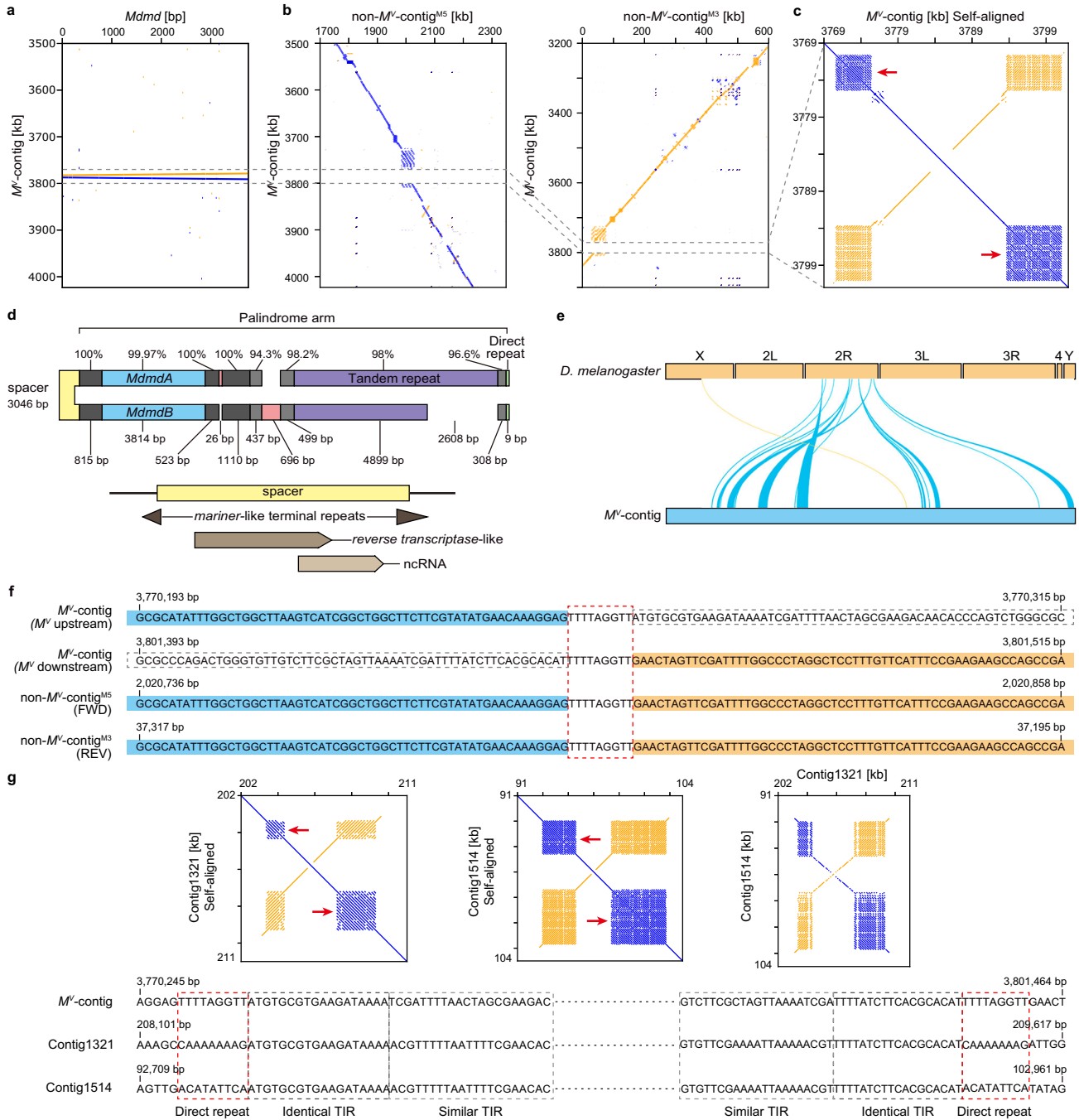

**Fig. 3 | The genomic structure and transposon-like signatures of $M^V$. a** Dotplot visualizations of *Mdmd* presence in inverse orientation on $M^V$-contig indicated by blue (forward) and orange (reverse) lines that represent longer stretches of sequence similarity. **b** Alignments between $M^V$-contig and non-$M^V$-contigs show $M^V$ was inserted in a tandem repetitive region, indicated by the gap. **c** Self-alignment of $M^V$ demonstrates that the main part of $M^V$ is a palindrome with a non-palindromic spacer sequence of 3046 bp in the middle. Two blocks of tandem repeats exist on each end of $M^V$ (indicated by red arrows). **d** Schematic drawing indicates the sequence contents of $M^V$ and percentage identities between palindromic arms. The spacer sequence shows homology to a reverse transcriptase sequence and an ncRNA. The red blocks and the missing parts represent insertions/deletions. The small green blocks indicate the existence of 9 bp direct repeats at the $M^V$ borders. **e** Single-copy

BUSCOs in the $M^V$-contig mainly correspond to those on chromosome 2 R (Muller element C, chromosome V in the housefly) in *Drosophila melanogaster*. **f** The 9-bp direct repeats (TTTTAGGTT) are found with one copy in non-$M^V$-contigs at insertion sites, indicated by the red dashed-line box. In non-$M^V$-contigs, the upstream and downstream sequences of TTTTAGGTT match with the upstream and downstream sequence of $M^V$ (indicated by the blue and orange shading). **g** Two examples of other genomic regions that contain the same tandem repeat blocks as $M^V$. The self-alignment figures demonstrate $M^V$-like palindromic structures in Contig1514 and Contig1321. Alignment between Contig1514 and Contig1321 shows sequence similarity only for the palindromic region but not for the palindrome flanking sequences. The TIRs in $M^V$ are also found in Contig1514 and Contig1321 palindromes, and direct repeats with the same 9-bp length are flanking them.

In all three investigated loci, $M^{III}$, $M^V$ and $M^Y$, homology is observed for upstream and downstream sequences of the intact *Mdmd* gene (Fig. 4f). The *Mdmd* flanking region described as the spacer in $M^V$, which includes the partial ncRNA, reverse transcriptase-like sequences and

mariner-like terminal repeats, is found in all three *M*-loci (Fig. 4f). Interestingly, $M^Y$ harbors sequence arrangements that are found in $M^V$ but not in $M^{III}$. In $M^V$, a block of tandem repeats is located downstream (~4 kb apart) of the intact *Mdmd*. Similar arrangements of the same

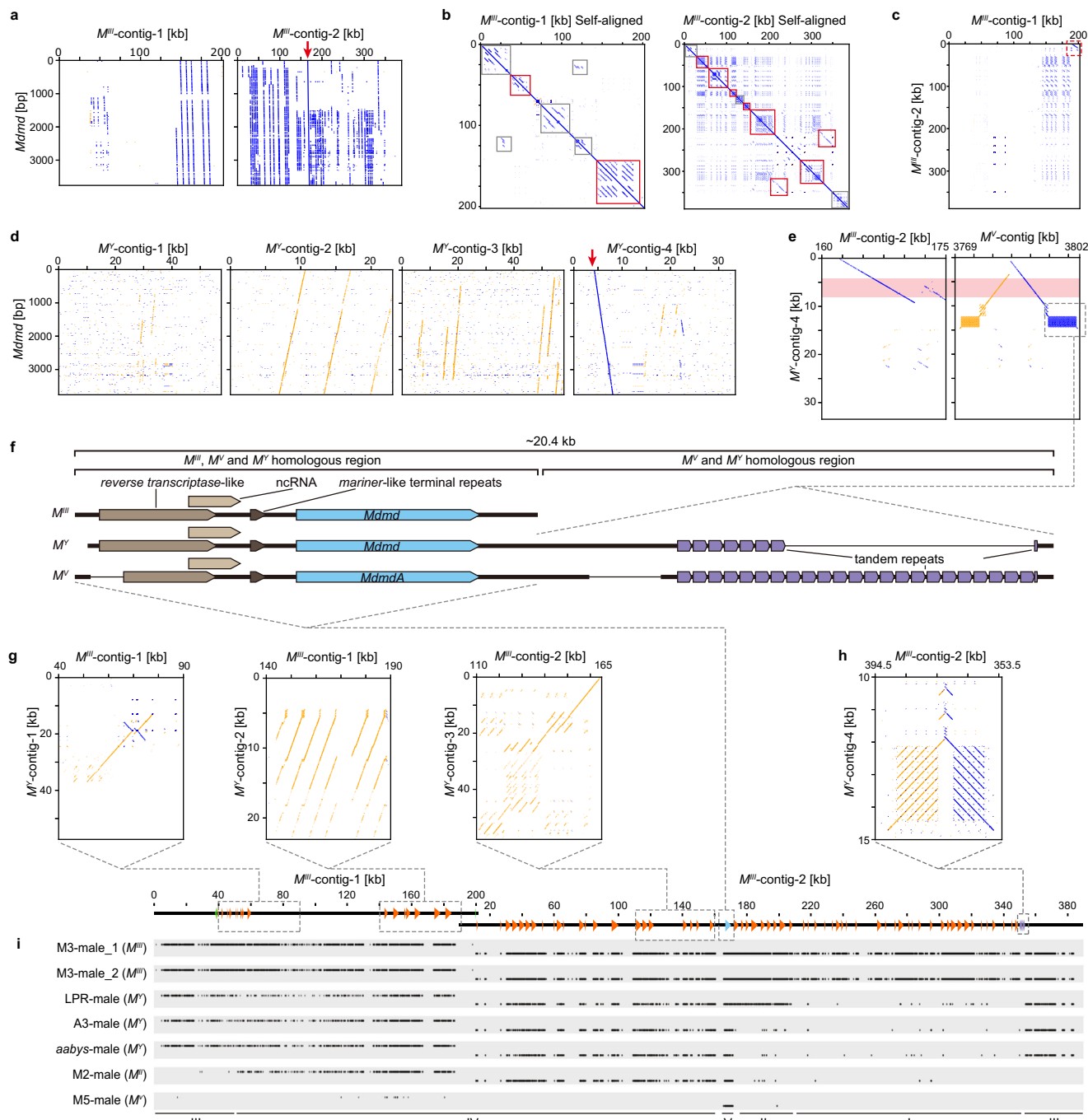

**Fig. 4 | Genomic structure of $M^{III}$, $M^Y$ and in comparison to $M^{II}$, $M^V$ loci.**
**a** Visualization of *Mdmd* sequences distribution in $M^{III}$-contigs. Only one complete *Mdmd* copy is found (blue line indicated by the red arrow). **b** Self-alignments of $M^{III}$-contigs show many tandem duplications clustering together (short blue lines, indicated by solid-lined boxes). Most of the duplications are *Mdmd* copies and their flanking sequences (red solid-lined boxes). Duplications of non-*Mdmd* sequences also exist (indicated by gray solid-lined boxes). **c** The end part of $M^{III}$-contig-1 shares homology to the beginning part of $M^{III}$-contig-2 (indicated by the red dashed-lined box). **d** Visualization of *Mdmd* sequences distribution in $M^Y$-contigs. Only one complete *Mdmd* copy is found in $M^Y$-contig-4 (blue line indicated by the red arrow). **e** $M^Y$-contig-4 show homology to regions of $M^{II}$- and $M^V$-contigs that contain intact

*Mdmd* gene which is indicated by red shading. **f** Schematic drawing of homologous regions that contain intact *Mdmd* in $M^{III}$, $M^V$ and $M^Y$. $M^Y$ and $M^Y$ both have a block of tandem repeats is located downstream (~4 kb apart) of the intact *Mdmd*. **g** $M^Y$-contigs show homology to various parts of $M^{III}$-contigs. **h** Tandem repeats that are adjacent to intact *Mdmd* in $M^V$ and $M^Y$ are also found in $M^{III}$, but exist near one end of $M^{III}$-contig-2. **i** The coverage for $M^{III}$-specific regions in various male genomic datasets that contain sequences of $M^{II}$, $M^{III}$, $M^V$, and $M^Y$ loci respectively. Schematic drawing shows *Mdmd* distribution (complete copy blue, truncated copies orange) in $M^{III}$-contigs. Other *M*-loci show various similarities to the $M^{III}$. I: Specific to $M^{III}$; II: Shared between $M^{III}$ and LPR $M^Y$; III: Shared among $M^{III}$ and all three $M^Y$; IV: Shared among $M^{II}$, $M^{III}$ and all $M^Y$; V: Shared among all tested *M*-loci.

repeats exist on both palindrome arms of $M^V$ (Fig. 4e, f). Although these repeats are also found in $M^{III}$, they exist near one end of $M^{III}$-contig-2 and are not adjacent to the intact *Mdmd* copy (Fig. 4h). Thus, $M^Y$ has a $M^{III}$-like structure but also shares sequence characteristics with $M^V$.

**Structures of $M^{II}$ and $M^Y$ based on Illumina read mapping**
Given the distinct architectures of $M^{II}$, $M^V$, and $M^Y$, we further examined the structures of the *M*-loci by mapping Illumina reads originating from males against the $M^{III}$-contigs. As Illumina reads are short and $M^{III}$

contigs contain many repetitive sequences, we first mapped female reads against the $M^{III}$-contigs to identify the repetitive regions in $M^{III}$ that are not $M$-locus-specific (Supplementary Fig. 5). When subsequently mapping the male reads, we subtracted the regions that showed female coverage. Thus, coverages only for $M^{III}$-locus-specific regions were retained. The Illumina reads from M3 males cover the entire $M^{III}$-locus, whereas sequences from male M5 cover only the region of the functional *Mdmd* (Fig. 4e). The coverage patterns of the three MY datasets are quite similar, though LPR-male showed additional coverage of $M^{III}$-specific sequences containing *Mdmd* copies (section II in Fig. 4e). This is consistent with the aforementioned results that the $M^{Y}$-locus in LPR male contains more *Mdmd* copies than the other $M^{Y}$-loci. M2-male sequences, for which we did not produce any PacBio sequencing data, show less coverage of $M^{III}$ than $M^{Y}$ but still cover about 300 kb of the $M^{III}$-locus (roughly 50%). The similar coverage patterns of $M^{I}$, $M^{III}$, and $M^{Y}$ datasets indicate the presence of highly similar sequence regions in these $M$-loci, which suggests an already complex architecture prior to the origin of these individual loci. Additionally, differences observed among $M^{Y}$ imply that the duplication events within the $M$-loci happened gradually in sequential steps, or that there was a large ancestral $M^{Y}$-locus, which was degraded differently in various strains.

### Sequence divergence of intact *Mdmd* copies in different *M*-loci

To examine sequence divergence across in intact *Mdmd* copies of $M^{I}$, $M^{III}$, $M^{V}$ and $M^{Y}$, we drew the *Mdmd* consensus sequence for the ORF based on our data on $M^{III}$ and $M^{V}$ as well as previously published sequences[13]. *Mdmd* in $M^{III}$ contains the highest number (8) of nucleotide differences from the consensus, whereas the two copies in $M^{V}$ have the fewest (1 in *MdmdA*, 2 in *MdmdB*, Supplementary Table 5). The number of divergent sites in *Mdmd* in $M^{I}$ and $M^{Y}$ is 4 and 6 respectively. Besides one divergent site that is shared between $M^{V}$ *Mdmd* copies (nucleotide position 527, G), different *Mdmd* sequences all possess unique sets of divergent sites, indicating these mutations arose independently in those $M$-loci. These few nucleotide differences did not allow for a reliable reconstruction of the ancestry of the *Mdmd* gene when using *Mdncm* as an outgroup.

### Chromosomal localization of *M*-loci and *M*-associated repeats on the X and Y chromosome

Alongside the aforementioned *Mdmd*-specific probe, we applied another probe, referred as the Mix probe, for FISH. The Mix probe was generated from PCR products that were specifically amplified from the genomic DNA of $M^{III}$-locus. The PCR products contains *Mdmd* fragments and intervening non-*Mdmd* sequences within truncated *Mdmd* copies in $M^{III}$. The Mix probe localized to the $M^{III}$-locus in male samples from the M3 strain as expected. Interestingly, two additional large signals at both ends of the X chromosomes that do not carry $M$-loci were observed in both male and female samples of the M3 strain (Fig. 2e, h). We further tested the Mix probe with samples from strains *aabys* (males with $M^{Y}$), M5 (males with $M^{V}$), and two Spanish strains (SPA1 and SPA4 in which samples possess $M^{X}$). The large signals on the X chromosomes were also detected in female *aabys*, and in both M5 female and male samples (Fig. 2f, g, i). When using the Mix probe on male samples from the *aabys* strain that carry a Y chromosome, we observed similar hybridization signals on the non-$M$-possessing X chromosome (Fig. 2d). In addition, a Mix probe hybridized region, much larger than the *Mdmd*-specific signal, covers almost the entire short arm of the Y chromosome. As the $M^{Y}$-specific signal cannot be distinguished from additionally detected Y chromosome regions, it indicates they are either overlapping or closely located. In SPA1 and SPA4 samples, the $M^{X}$-specific signal also cannot be distinguished from additionally detected X chromosome regions (Supplementary Fig. 1i, j).

The Mix probe likely detected large terminal regions of the sex chromosomes by hybridizing to repeats shared by the XY

chromosomes. We refer to these repeats as *M* And Sex chromosome located (MAS) repeats. Note that those detected terminal regions likely contain sequences other than MAS repeats as well. The MAS repeats seem to be universally present on the X and Y chromosomes regardless of the strain of origin and the presence or absence of *M*. However, $M^{III}$ also contains MAS repeats as the Mix probe originated from $M^{III}$ genomic DNA.

## Discussion

In order to compare genomic structures of *Mdmd*-containing loci, we assembled three male housefly genomes, one with the male-determining locus on chromosome V ($M^{V}$), one on chromosome III ($M^{III}$), and one on the Y chromosome ($M^{Y}$). The *M*-loci differ considerably in size, sequence composition and structure. $M^{V}$ is the simplest and contains only two intact *Mdmd* copies with minimal sequences in between. In contrast, $M^{III}$ and likely $M^{Y}$ contain a single complete *Mdmd* ORF, many truncated *Mdmd* copies, and other pseudogenes that are absent in $M^{V}$. The male-determining capacity of $M^{V}$ demonstrates the importance of the intact *Mdmd* copy as the male-determining factor, and suggests that the remaining sequences in other *M*-loci are dispensable for sex determination.

We find that the *Mdmd* gene can be embedded in very different genomic regions on the chromosomes on which it is located. Palindromes are often found in regions on Y/W sex chromosomes that contain genes with sex-determining function, such as the Y chromosome of mammals[24–28], European rabbits[29], and the W chromosome of the white-throated sparrow[30], where they appear to facilitate concerted evolution via gene conversion[24,29]. The palindromic $M^{V}$ has the fewest nucleotide changes between the two *Mdmd* copies and the *Mdmd* consensus may reflect a similar process of concerted evolution. In contrast, *Mdmd* on chromosome III points to a very different genomic dynamic. $M^{III}$ contains many repeated but degenerated sequences, which is consistent with the model of junk DNA and mutational degeneration of sex-determining loci[6]. At this stage, we have no evidence for a functional role of the duplicated sequences and the *Mdmd* truncated copies.

Different mechanisms likely underlie the formation of distinctive *M* architectures. $M^{V}$ seems to have been produced by transposase activity as we found sequence signatures of TIRs and DRs as well as highly similar palindromic structures in other regions with the same TIRs. As the TIRs are not present in the corresponding non-$M^{V}$-contigs, their insertion likely has occurred together with the *Mdmd* gene(s). An intriguing possibility is that the TIRs are involved in translocation of *Mdmd*, and $M^{V}$ potentially gained the ability to change its genomic location via nonautonomous translocation mediated by the TIRs. Other genomic processes may be responsible for the complex architecture of $M^{III}$. The multiple *Mdmd* copies in $M^{III}$ could be the result of double-strand breakage and homologous repair that are known to generate tandem duplications[31]. According to the duplication-dependent strand annealing model[31], several traces are characteristic of such duplications, i.e., microhomology in template and duplicated sequences that allow reinvasion during homology repair. Upon examination of duplicated sequences in $M^{III}$ contigs we indeed found such signatures (Supplementary Discussion, Supplementary Fig. 6) supporting the occurrence of these genomic processes.

There is another process that may have affected the genomic evolution of *M*-loci. Populations with multiple *M* loci often carry the *Mdtra* gene variant, $Mdtra^{D}$, which could have also affected *M* structure. $Mdtra^{D}$ is epistatically dominant over *M* and can promote female development even in the presence of *M*, which allows females to carry *M*-loci[8,32]. Although recombination in males may be reduced, in females it is not. Hence, if a female is homozygous for an *M*-locus, unequal crossover within the *M*-locus may cause expansion or reduction of *M*, although we currently do not have evidence to support this hypothesis. In addition, it is presently unknown if the $Mdtra^{D}$ allele

originated before, during, or after the evolution of the various *M*-loci, which could affect the plausibility of this model.

From our results, we infer the complex structure of *M* formed via gradual duplication on the Y chromosome. During this process, not only did the copy number of *Mdmd* increase, but the *M*-surrounding sequences likely also became a part of *M*, by getting intercalated by *Mdmd* copies. Thus, non-*Mdmd* sequences in *M*[^Y] are expected to show homology to the X chromosome. We tested this hypothesis by mapping Pacbio reads of *aabys*-male and M5-male to the *M*[^Y]-contigs. Indeed, we observed some regions intervening *Mdmd* copies showed coverage to non-*M*[^Y] reads (Supplementary Fig. 7). *M* And Sex chromosome located (MAS) repeats detected by the Mix probe indicate abundant repetitive sequences that are prevalent on the X and Y chromosomes but not on other chromosomes. This is consistent with previous reports[21,33] that housefly XY chromosomes mostly consist of highly repetitive sequences that are unique to them. *M*[^III] also contains MAS repeats given the fact that the Mix probe was derived from *M*[^III] DNA but there are not MAS repeats on the non-*M* third chromosome. This suggests that *M*[^III] has originated from the translocation of a DNA segment from the Y chromosome that contained *Mdmd* and MAS repeats.

*M*[^Y] contains sequences characteristic shared with both *M*[^III] and *M*[^V], which points to two possible evolutionary routes for how the housefly *M*-loci arose. The first is that *Mdmd* originated on chromosome V and then translocated to an X chromosome (according to current karyotype numbering), which converted the X into what we refer to as the housefly Y chromosome now. *M* on the Y then evolved a complex, tandem duplicated structure, which later translocated to other chromosomes. A second possibility is that *M* first established on the Y, and subsequently translocated to other chromosomes embedded in DNA fragments that vary in size, ranging from a single copy when transposing to chromosome V to many copies when transposing to chromosome III. Although the first scenario appears more intuitive as it follows a "simple to complex" order, we consider the second scenario more plausible for several reasons. Based on cytogenetic data the XY is the only morphologically differentiated chromosome pair (Y being smaller than the X chromosome), whereas other *M*-carrying chromosomes (e.g., *M*[^III] and *M*[^V]) do not visibly differ from their non-*M*-carrying counterparts. This observation is consistent with the hypothesis that the Y chromosome is "older" than other *M*-carrying chromosomes, whereas the other proto-Y chromosomes have not experienced substantial degeneration and remained intact. An indication for the "young" status of *M*[^V] is that high sequence identity is observed between homologous sequences of non-*M*[^V]-contigs and *M*[^V]-contigs except for the small *M*[^V]-locus region. This suggests a minimal degree of divergence for the *M*-surrounding regions on the chromosome V pair. With our current data, we cannot entirely discern the evolutionary histories of the various *M* loci as we lack information regarding the X chromosome region that is homologous to the *M*[^Y] locus. This would require constructing chromosomal-level assemblies of different *M*-carrying chromosomes and comparing the sequence divergence and recombination rates between *M*-carrying chromosomes and their non-*M*-carrying homologs in future studies.

Our study sheds light on the complex evolution of the polymorphic sex determination system of the housefly. *Mdmd* originated as a copy of the *Mdncm* gene, which was followed by duplication events generating multiple, incomplete *Mdmd* copies[13]. Our results imply the intact *Mdmd* was translocated from one chromosome to the others embedded in large DNA fragments which varied in size and often contained incomplete *Mdmd* copies. Transposable elements were likely involved in the translocation events, such as the establishment of *M*[^V] resulting in a distinctive palindromic structure. An interesting finding is that even *M*-loci with comparable structures (*M*[^II], *M*[^III] and *M*[^V]) show signs of diversification. For example, *M*[^III] contains specific genomic regions that are not found in other investigated *M*-

loci. Even *M*[^Y] from different populations vary in *M*-locus sequence, indicating that *M*-loci are independently evolving in separate populations. In summary, our study demonstrates that nascent sex determination regions can be subject to different genomic processes leading to diverse genomic architectures.

## Methods

### Data type and housefly strains used in each analysis are listed in Table 1.
Notably, males and females in M2, M3 and M5 strains have two X chromosomes and no Y chromosomes. Strains established from collections of wild populations contain various combinations of *M*[^I], *M*[^II], *M*[^III], *M*[^Y], or *M*[^X]. Both hemizygous and homozygous *M*-loci were found in these strains.

### Analysis of Mdmd copy number variation
The copy number of *Mdmd* in different *M*-loci was determined by mapping the raw reads to the published *Mdmd* sequence[13] (Accession: KY020049.1). The mapping and coverage analysis was done with Burrows-Wheeler Aligner[34] mem (BWA, v0.7.17) and SAMtools[35,36] (v1.10) using the default parameters. The average coverage of *Mdmd* was calculated for each base. The *Mdmd* coverage was standardized among datasets by calculating a relative coverage which is a ratio of *Mdmd* coverage dividing by the coverage of a single copy autosomal reference gene. To minimize potential errors, three reference genes were selected, i.e., *Mdtra*[35] (Accession: GU070694.1), *yellow* (*MdY*, Accession: KY979110.1) and *asense* (*Mdase*, Accession: XM_005176302.3). The final *Mdmd* copy numbers for each dataset were calculated by taking the average relative coverage for each of the three reference genes and multiplying by two as autosomal genes have two alleles but the *M*-locus is hemizygous. Sequence depth files that are generated by SAMtools and are used to calculate coverages are provided as a Source Data file.

### Chromosome preparations
Chromosome slides were prepared from the brain tissues of third instar larvae. Spreads of mitotic chromosomes were made according to the method of ref. 37. with slight modification. In short, larval brains were dissected in Ringer's solution, pre-treated in hypotonic solution (75 mM KCl) for 10 min and then fixed in Carnoy's fixative (ethanol:acetic acid, 3:1) for 10 min. Fixed tissues were then transferred to glass slides (Thermo Fisher Scientific SuperFrost Microscope Slides) with a drop of 60% acetic acid and spread with a tungsten needle on a 45 °C heating plate. Slides were examined under a phase contrast microscope (Carl Zeiss Axio Lab.A1) to check whether the nuclei were appropriately spread before FISH.

The remaining larval tissue was used for DNA extraction using a high salt protocol[38] followed by PCR with primers (Mdmd_1as, GATTGGCTCAGATCGGCGTA and Mdmd_6as, GGTTGACGCGGA CAATCAAC) designed on *Mdmd* specific sequences according to ref. 13. to determine whether the larva possessed the *Mdmd* sequences. PCR was conducted with Platinum II *Taq* Hot-Start DNA Polymerase (Thermo Fisher Scientific) according to the manufacturer's instructions. The thermocycling program was as follows: initial denaturation at 94 °C for 2 min; 30 cycles of 15-s denaturation at 94 °C, 15-s annealing at 60 °C, 1 min 15-s extension at 72 °C, with a final extension of 3 min at 72 °C. PCR products were visualized on a 1% agarose gel in TAE buffer to evaluate the presence of *Mdmd* in the samples.

### Probe preparation
To prepare probes for FISH experiments, DNA fragments of *M*[^III] were amplified with *Mdmd*-specific primer pair Mdmd_FISHs, GGAAGTCG TATTGGAAGTAGT and Mdmd_FISHa, ATTTGGTGCGCCCTTCT using Platinum II *Taq* Hot-Start DNA Polymerase according to the manufacturer's instructions. The PCR product contains a mixture of *Mdmd* fragments and non-*Mdmd* fragments as many intergenic sequences

exist in $M^{III}$ such as repeats and transposable elements. A Mix probe was prepared directly from purified PCR products by labeling with digoxigenin (DIG)−11-deoxyuridine triphosphate (dUTP) using the DIG-Nick Translation Mix (Roche) according to the manufacturer's instructions. Labeled in the same way, an *Mdmd*-specific probe was made from a cloned *Mdmd* gene which sequence was confirmed by Sanger sequencing.

## Fluorescence in situ hybridization

The FISH procedure was adapted from ref. 39. with minor modifications. Chromosome slides were pretreated with 100 μg/ml RNase A in 1×PBS for 1 h at 37 °C, followed by washing three times with 2 × SSC at room temperature for 5 min each. Subsequently, the slides were denatured in 2 × SSC containing 70% formamide at 68 °C for 3.5 min, dehydrated by passing them through an ice-cold ethanol series (70%, 90%, 100%; 5 min each) and air-dried. The 20 μl probe mixture contained 200–300 ng digoxigenin-labeled DNA probe, 50% (v/v) deionized formamide, 10% (v/v) dextran sulfate in 2 × SSC. The probe was denatured at 90 °C for 5 min and rapidly cooled on ice for 10 min. The denatured probe mixture was then applied to the slides and left to hybridize at 37 °C for at least 14 h.

After hybridization, slides were washed with 2 × SSC, 50% formamide at 42 °C for 10 min, followed by three washes with 2 × SSC at 42 °C for 5 min each. Slides were blocked with 3% (w/v) bovine serum albumin blocking buffer (dissolved in 4 × SSC with 0.1% Tween 20). Probes were detected with Anti-Digoxigenin-Rhodamine (Roche) by incubating at 37 °C for an hour. Slides were then washed three times with washing buffer (4 × SSC with 0.1% Tween 20) at 37 °C for 5 min each. After washing, slides were shortly rinsed with 2 × SSC and air-dried. Chromosomes were counterstained with ProLong Diamond Antifade Mountant with DAPI (Thermo Fisher Scientific). Signals were detected with a Leica epifluorescence microscope (DMI6000 B) equipped with a Leica CCD camera (DFC365 FX) and analyzed with Leica Application Suite X (3.4.2.18368.1.2). Chromosomes were numbered according to ref. 33. Signals were only considered as a successful hybridization if they were observed with consistent chromosomal locations on at least 20 metaphase nuclei.

## Genome sequencing

We employed Pacbio sequencing to generate datasets for genome assembly. Two datasets were generated for M3 genome assembly. For Dataset1, 25 adult males from a single pair of parents were pooled for DNA extraction using Genomic-tip 100/G (Qiagen) according to the instruction manual. A DNA library was constructed with SMRTbell at the Leiden Genome Technology Center in the Netherlands and Blue-Pippin was used for size selection of >10 kb fragments. For Dataset2, 20 non-related adult males from the M3 strain were pooled for DNA extractions using Nucleo Bond AXG columns (Macherey Nagel) according to the instruction manual. A DNA library was constructed at the Functional Genome Center Zürich (FGCZ), Switzerland and >20 kb fragments were selected for sequencing. Two datasets together correspond to an approximate genome sequencing coverage of -116× (-84× for Dataset1 and -32× for Dataset2). For the M5 genome, genomic DNA of 3 adult M5 males was pooled and extracted using Nucleo Bond AXG columns (Macherey Nagel) according to the instruction manual. A DNA library was constructed at the FGCZ and sequenced on three Pacbio Sequel IIe cells generating HiFi reads, with an approximate coverage of -161×. For the *aabys*-male genome, DNA was extracted from flash frozen house fly male heads using the Qiagen Blood and Cell culture DNA MIDI Kit. High molecular weight DNA extraction was prepared for input of PacBio library prep according to ref. 40. A DNA library was constructed at the Clemson University Genomics and Computational Biology Facility (Clemson, SC, USA) and sequenced on Pacbio RSII cells using P6-C4 chemistry, generating a dataset with an approximate coverage of -13×.

For comparing genomic sequences of different *M-loci*, sequence data were obtained at the FGCZ, on the Illumina HiSeq2500 platform, generating 101 bp paired-end reads for M2 males and M5 males, 126 bp paired-end reads for M3 males or 151 bp paired-end reads for M3 females. All the genomic DNA was isolated using NucleoBond AXG 20 (MACHEREY-NAGEL) according to the instruction manual. For each DNA sample, the DNA was extracted from a pool of 5 flies. Separate libraries were prepared from each pool of DNA. The genomic sequences for $M^Y$, published in ref. 21., were downloaded from the NCBI database.

## Genome assembly

We performed M3 genome assembly using Canu[41] (v1.8) with the following parameter settings: corOvlErrorRate = 0.24, obtOvlErrorRate = 0.045, utgOvlErrorRate = 0.045, corErrorRate = 0.3, obtErrorRate = 0.045, utgErrorRate = 0.045, cnsErrorRate = 0.075, genomeSize = 900,000,000. The assembled M3 genome was then error-corrected with Quiver[42] (v2.2.1). The M5 genome was assembled using a newer Canu[43] version (v2.2) but with the same settings as for M3. A Quiver correction was not necessary, because Pacbio HiFi reads were used. The *aabys*-male genome was assembled using Flye[44] (v2.9.3) with the parameter "--no-alt-contigs". Male illumina reads of the same strain were then used to polish the genome assembly with Pilon[45] (v1.24). The summary statistics of the assembled genomes were obtained with QUAST[46] (v4.6.3). BUSCO[47] (v5.0.0) was used to estimate the completeness of the genomes by estimating the percentage of assembled universal protein-coding genes in dipteran lineages. Furthermore, the repeat content including interspersed repeats and tandem repeats of the M3 genome was analyzed with RepeatModeler (v1.0.11) and RepeatMasker (v4.0)[48].

## Genomic analysis of $M^{III}$, $M^V$ and $M^Y$

To identify sequences spanning the $M$ region, we performed a search with BLAST using the published *Mdmd* sequence[13] against our newly assembled genomes and identified *Mdmd*-containing contigs. To exclude *Mdncm* sequences that shared a significant degree of sequence similarity with *Mdmd*, contigs that contain a single-copy sequence with over 95% identity to *Mdncm* were removed from the *Mdmd* contig pool. Non-$M^V$-contigs were identified by using $M^V$-contigs to search for sequence similarity against the genomes by BLAST. Synteny analysis was based on single-copy BUSCO information in $M^V$-contig and *D. melanogaster* and plotted by R package RIdeogram[49]. TIRs and DR were manually checked based on alignments.

All the *Mdmd* sequences in $M^{III}$-contigs were manually checked based on the previous BLAST search and grouped into different *Mdmd* copies based on the sequence continuity and position on the contigs. $M^{III}$-contigs were screened for annotated sequences by using BLASTto compare the contigs against the NCBI *Musca domestica* (Taxid: 7370) Nucleotide Collection database. Obtained hits of annotated genes were subsequently used to search for the presence of these genes on other contigs in the M3 genome by BLAST.

Dotplot visualization of the alignments were done via Flexidot[50] (v1.06) with different wordsize setting, i.e., 15 for alignments of *Mdmd* sequences in $M^V$-contig and $M^{III}$-contigs, 10 for alignments of *Mdmd* sequences in $M^V$-contigs, 100 for alignments between $M^V$-contig and non-$M^V$-contigs, 20 for alignments between $M^{III}$-contigs and $M^Y$-contigs, 50 for the rest.

## Analysis of $M$-locus coverage

The female Illumina reads were mapped to $M^V$-contigs and $M^{III}$-contigs to detect sequences that are specific to $M^V$ and $M^{III}$ using BWA mem with adjusted parameters, i.e., -t 16 -M -P -c 5000 -k 65 -B 7 -w 10 -d 60. The BWA output was used to calculate read depth for each nucleotide position using SAMtools with the "depth" function.

To detect sequence content variation between different $M$-loci, male Illumina reads were mapped to $M^{III}$-contigs using the same pipeline with BWA and SAMtools as described above. To minimize the

false positive alignments, a minimum coverage value of 5 was set when plotting with R package ggplot2[51] as dotplot.

Minimap2[52] (v2.26) was applied to map Pacbio raw reads of *aabys*-male and M5-male to $M^Y$-contigs. Integrative Genomics Viewer[53] (v2.17.4) was used to visualize the mapping results.

## Verification of *MdmdA* and *MdmdB* transcription

RNA and DNA were simultaneously isolated from individual M5 pupae using TRIzol reagent (Thermo Fisher Scientific). RNA was isolated according to the manufacturer's instructions whereas gDNA was isolated from the organic phase using a back extraction protocol as described in ref. 54. RNA samples were DNase-treated with the Invitrogen TURBO DNA-*free* kit (Thermo Fisher Scientific), and approximately 2 μg RNA was converted to cDNA using the RevertAid First Strand cDNA Synthesis Kit (Thermo Fisher Scientific) with the oligo(dT)$_{18}$ primer included in the kit in a total reaction volume of 20 μL. The sex of the samples (using gDNA template) as well as the transcription of *MdmdA* and *MdmdB* (using cDNA template) were tested using primers that amplify a region of *Mdmd* that includes the intron and the SNP between *MdmdA* and *MdmdB*, Mdmd_4F (TTGCATCAAGGCAAGTTGGA) and Mdmd_4R (TCTGAATCACTTGAA-GAATCGT). PCR was carried out in 20 μL reaction volumes consisting of 1× DreamTaq buffer, 0.2 mM dNTPs, 0.2 μM of each primer, 0.5 U DreamTaq DNA polymerase (Thermo Fisher Scientific) and 1 μL 10× diluted cDNA (or 50–100 μg gDNA). The thermocycling program was as follows: initial denaturation at 94 °C for 3 min; 35 cycles of 30 s denaturation at 94 °C, 30 s annealing at 59 °C, 1 min 15 s extension at 72 °C, with a final extension of 3 min at 72 °C. Amplification was verified by gel electrophoresis using a 1% agarose gel in 1× TAE buffer. To remove contaminants before sequencing, 5 μL of the amplified samples was combined with 1.6 U exonuclease I and 0.12 U FastAP (both Thermo Fisher Scientific) in a total volume of 9 μL, and incubated at 37 °C for 30 min. The reactions were inactivated at 80 °C for 15 min after which the samples were sent for Sanger sequencing (Eurofins).

## Reporting summary

Further information on research design is available in the Nature Portfolio Reporting Summary linked to this article.

## Data availability

All genomic data supporting the findings of this study are available under BioProject: PRJNA1013067 and PRJNA1072234 in the NCBI database. The M3, M5, and *aabys*-male assemblies were deposited at NCBI GenBank under accession JAVQME000000000, JAVVNY000000000, and JAZGUT000000000, respectively. Illumina reads generated from this study were deposited at NCBI Sequence Read Archive (SRA) under accession numbers: SRX21801162 (M2-male), SRX21801164 (M3-male_1), SRX21801165 (M3-male_2), SRX21801166 (M3-female_1), SRX21801167 (M3-female_2), SRX21801163 (M5-male). Illumina reads of MY samples were from the previous publication[21] and were downloaded from SRA (accession numbers: SRX2154714- SRX2154719). Reference gene sequences, *Mdmd* (accession: KY020049.1), *Mdtra* (accession: GU070694.1), *MdY* (accession: KY979110.1) and *Mdase* (accession: XM_005176302.3), were obtained from GeneBank. Source data are provided as a Source Data file. Source data are provided with this paper.

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

## Acknowledgements

We thank Anna Rensink, Peter Hoitinga, Ljubinka Francuski Marcetic, Marloes van Leussen, Dré Kampfraath, Jan Keijser, Jacopo Martelossi, and Alexander Suh for their advice and assistance. We thank the Center for Information Technology of the University of Groningen for their support and for providing access to the Hábrók high performance computing cluster. This work was completed in part with resources provided by the Research Computing Data Core at the University of Houston. X.L. is supported by China Scholarship Council Scholarship no. 201606330077.

## Author contributions

X.L., L.W.B., D.B., L.v.d.Z., and E.W. designed the project; L.W.B., D.B., L.v.d.Z., E.W., and R.P.M. supervised and funded the project; X.L., S.V., Y.W., and F.M. performed molecular experiments and analyzed data; X.L., S.V., J.H.S., E.G., E.N.K., S.Y.M., M.P., S.A., M.A.S., M.D.R., and R.P.M. contributed to genomic data collection, genome assembly and genomic analysis. X.L., S.V., L.W.B., D.B., L.v.d.Z., and E.W. designed the figures in the manuscript. The manuscript was written by X.L., S.V., L.W.B., D.B., L.v.d.Z., and E.W. All authors reviewed the paper.

## Funding

## Competing interests

The authors declare no competing interests.
