## [Transparent Peer Review file · Nature Communications]

Divergent evolution of male-determining loci on proto-Y chromosomes of the housefly

Corresponding Author: Dr Xuan Li

Version 0:

Reviewer comments:

Reviewer #1

(Remarks to the Author)

The evolution of sex determination loci is one of the most intriguing themes in biology, and the housefly serves as an excellent model for this purpose. The authors attempt to elucidate the evolutionary process of the male determination gene locus, referred to as "M," in housefly, which has shown diversity in terms of the number and position of M loci within the species, through extensive genome sequence analysis and FISH. While the experimental results obtained are highly interesting, they still appear insufficient to fully support the model proposed by the authors.

If the authors' hypothesis that "M3-like Ms originated from the Y chromosome" is correct, then MASX on the X chromosome does not contain Mdmd, including pseudo-genes, and only MASX on the Y chromosome does contain Mdmd. Furthermore, in MASX, it is presumed that Mdmd initially arose within one repeat and then spread to the surrounding repeats through concerted evolution, so there should be repeats similar to MASX (not containing Mdmd) outside of the repeats containing Mdmd. While these discussions are implied by the absence of signals on the X chromosome in FISH using the Mdmd-specific probe and the observation of the mix probe signals over a broader range on the Y chromosome than the Mdmd-specific probe signals, they are not mentioned in the paper. The authors have analyzed the sequences of the M3 and M5 lineages in this study, but to substantiate their theory, I believe it is essential to analyze the genome of the MY lineage, particularly through comparative analysis of the sequences of MASX and MASX.

It seems natural to consider that the dominant female-determining factor, "traD," has also played a role in the diversification of Ms. However, this paper does not make any reference to this aspect. As indicated in Table 1, there are populations of this species with multiple Ms. While acquisition of additional M locus could potentially trigger the evolution of new sex chromosomes, as well as chromosomal translocations, having multiple dominant male-determining loci results in an increasingly male-biased sex ratio, making the establishment of multi-locus Ms challenging in typical cases. Nonetheless, traD can mitigate male bias, making it possible for multi-locus Ms to arise. Therefore, it is also necessary to investigate the variant patterns of tra in the lineages used to further explore these considerations.

Reviewer #2

(Remarks to the Author)

Musca domestica presents an intriguing variability of the chromosomal locations of the M region. The authors investigated various strains at a genome-wide level with differently located M-loci, which share a conserved Mdmd gene promoting male sex determination. Taking advantage of DNA long reads sequencing technologies, the authors produced massive sequence data and related analyses showing that all M-loci also have different molecular architectures and confirmed different cytogenetic localization in situ. By comparing the various genomic data, the authors reconstructed the potential evolutionary trajectory of the various M-loci. They concluded that they arose from unique independent translocations of large DNA fragments containing Mdmd from an ancient Y chromosome, leading to diverse M-loci genomic architectures. For example, the authors found convincing indications that MV was likely produced by transposase activity, while MIII was produced by double-strand breakage and homologous repair, typical of tandem duplications. This work contributed to deepening our fundamental knowledge of the evolution of sex-determining primary signals at the genetic and chromosomal levels.

Minor suggestions follow.

Abstract

In the housefly, male-determining M-loci can reside on the Y chromosome

28 (MY), but also on the X chromosome (MX) and autosomes II (MII), III (MIII), V (MV). We investigated
29 differently located M-loci, all containing the primary male-determining gene *Musca domestica* male
30 determiner (Mdmd).

I suggest mentioning the *Musca Mdmd* master gene from the very start of the abstract and specifying if all Ms are Mdmd orthologues.

38 containing the same primary signal, independently diverged into regions of various complexity, leading
39 to distinctly different male-determining loci.

I suggest to mention that the same primary signal is Mdmd.

A specific prediction of the canonical sex
50 chromosome evolution model is that a sex-determining region will undergo progressive recombination
51 suppression to preserve its function^{3–7}.

Specify here briefly why the sex-determining region requires progressive recombination suppression to preserve its function.

Results

74 from Pacbio SMRT sequencing of the strains M3

Why have the authors sequenced M3 in addition to M5? The experimental design description could be more straightforward.

As there was only a female genome available²², we assembled male genomes
74 from Pacbio SMRT sequencing of the strains M3 (~116Å~ total coverage) and M5...
79 addition, we performed Illumina sequencing on strains M3 (males and females), M5 (males), and M2
80 (males, which carry MII).

Why have the authors sequenced by Illumina also M2 in addition to M3 and M5?

83 As a first estimate of sequence divergence between M loci we determined the coverage of Mdmd
Please specify the correlation between the coverage of Mdmd and sequence divergence. Needs to be clarified.

aabys⁸⁸

male, ~19.62; A3-male, ~19.74) were similar. Interestingly, one MY dataset, LPR-male, showed a higher
89 average coverage (~34.78) than the other two MY datasets.

What is LPR-male?

92 To identify the cytogenetic localization of M loci

I suggest adding the type of *Musca* tissues and cells the authors have used here.

98 not pericentromeric. Using samples from multiple laboratories
Which laboratories?

100 chromosomes, regardless of strain origin (supplementary fig. S1), indicating a single evolutionary origin
101 of each of these M-loci

I would substitute the term "indicating" with "suggesting."

105 The results above indicated that the complexity of MV was lower than that of the other M-loci
Please specify the meaning of the term complexity. Do the authors mean simply the copy number?

109 estimated coverage of Mdmd for the M5 genome. Only a single synonymous SNP, located in exon 2,
110 was found between these two Mdmd copies

SNP in ORF? Aminoacidic substitution observed?

.
The ~31 kb sequence,
118 which exists only on the MV-contig and includes the two opposing Mdmd copies, can thus be considered
119 as the complete MV locus. MV is integrated in a ~10 kb tandem repeat block shared between the MV-contig
120 and non-MV-contigs.

What is the sequence similarity of the 10Kb tandem repeat block shared between the homologous V chromosomes?
Approximate divergence time?

162 MIII are likely degenerated pseudo-copies. None of these genes have been reported to be involved in sex163
determination

Significantly few known genes involved in sex determination (hence as primary signals) have been isolated. These are
different in various groups of insects. Please specify and expand the sentence "None of these genes have been reported to
be involved in sex determination". Did the authors expect differently? In what sense?

185 the aforementioned results that the LPR male MY genome contains more Mdmd copies than the other
186 male MY genomes.
Please give more information on LPR male and MY male genomes (Strains? Individuals of different origin?).

189 origin of these individual loci. Additionally, differences observed among MY imply that the duplication
190 events on the M loci happened gradually and at different in frequencies.
Please correct "at different in frequencies."
Gradually? Or in subsequent steps?

216 examine whether the MASX can be separately detected from MX, we performed Mix probe-based FISH
217 on samples from two Spanish strains that possess an M-locus on the X chromosome. As no separate
218 signals are observed (supplementary fig. S1i, j), MX is either included in or closely located to MASX

Please define MX as soon as it is introduced in the text, not later. It is confusing.
Before describing these data, a brief recap of the various strains having the M in various locations could be helpful.

248 Different mechanisms likely underlie the formation of distinctive M architectures. MV seems to have been
249 created by transposase activity as we found sequence signatures of TIRs and DRs and highly similar

I would avoid using the term "created" and substitute it with produced, generated, etc....

Author Rebuttal letter:

We thank the reviewers for their valuable comments that have improved the
presentation of our results. Below we provide answers to the reviewers' comments. We hope that our
manuscript can now be accepted for publication in Nature Communications.

Reviewer #1:

The evolution of sex determination loci is one of the most intriguing themes in biology, and the housefly

serves as an excellent model for this purpose. The authors attempt to elucidate the evolutionary process of the male determination gene locus, referred to as "M," in housefly, which has shown diversity in terms of the number and position of M loci within the species, through extensive genome sequence analysis and FISH. While the experimental results obtained are highly interesting, they still appear insufficient to fully support the model proposed by the authors.

If the authors' hypothesis that "M3-like Ms originated from the Y chromosome" is correct, then MASX on the X chromosome does not contain Mdmd, including pseudo-genes, and only MASY on the Y chromosome does contain Mdmd. Furthermore, in MASY, it is presumed that Mdmd initially arose within one repeat and then spread to the surrounding repeats through concerted evolution, so there should be repeats similar to MASX (not containing Mdmd) outside of the repeats containing Mdmd. While these discussions are implied by the absence of signals on the X chromosome in FISH using the Mdmd-specific probe and the observation of the mix probe signals over a broader range on the Y chromosome than the Mdmd-specific probe signals, they are not mentioned in the paper. The authors have analyzed the sequences of the M3 and M5 lineages in this study, but to substantiate their theory, I believe it is essential to analyze the genome of the MY lineage, particularly through comparative analysis of the sequences of MASX and MASY.

>We thank the reviewer for insightful comments. We agree to better discuss the origin of the M loci and consider genomic sequences of MY in addition to MIII and MV. We have now collected genomic sequences of male samples from the aabys strain, which possess MY, and compared MY sequences to those of MIII and MV. We find that the majority of the MY sequences show homology to multiple regions of MIII-contigs, whereas the genomic region containing the intact Mdmd gene shows a sequence arrangement that is specific to MV. As MY possesses characteristics of both MIII and MV, we interpret this as support for the ancestry of MY. We include our new findings in the Results section (see lines 201-223, and updated figure 4). We comprehensively discuss these findings, together with other results and information from previous publications, and why we think ancestry of MY is the most parsimonious scenario (see lines 331-350). Moreover, we have attempted to create chromosome-level assembly of the X and the Y so that we could compare the genomic surroundings (i.e. MAS repeat sequences) of MY between the two chromosomes at a large scale. However, the XY assembly turns out to be poorly resolved, which currently prevents further analysis of the sequences of MASX and MASY.

As the new Y sequences were kindly provided by Jae Hak Son and Richard P. Meisel, and both contributed significantly to the manuscript revision, we have added them as co-authors.

It seems natural to consider that the dominant female-determining factor, "traD," has also played a role in the diversification of Ms. However, this paper does not make any reference to this aspect. As indicated in Table 1, there are populations of this species with multiple Ms. While acquisition of additional M locus could potentially trigger the evolution of new sex chromosomes, as well as chromosomal translocations, having multiple dominant male-determining loci results in an increasingly male-biased sex ratio, making the establishment of multi-locus Ms challenging in typical cases. Nonetheless, traD can mitigate male bias, making it possible for multi-locus Ms to arise. Therefore, it is also necessary to investigate the variant patterns of tra in the lineages used to further explore these considerations.

>Indeed, the dominant female-determining factor traD is relevant for the evolution of polymorphic M loci as traD may allow for homozygosity of an M locus and recombination. For example, MIII consists of many replicated sequences, unequal crossover between both copies may result in either expansion or reduction of the locus. This is one possible explanation for why MIII has a larger size than other M loci such as MII and MY. We added this discussion on the role of traD in lines 319-324.

It is indeed predicted that traD frequencies correlate with M frequencies in populations. Information on traD frequencies in i.e. Spanish populations used in the current study has been presented in our previous publication (Li et. al., Insect Science, 2021). Reporting population variation in traD and M frequencies goes beyond the scope of the current manuscript that focuses on M genomic structures.

Reviewer #2:

Musca domestica presents an intriguing variability of the chromosomal locations of the M region. The authors investigated various strains at a genome-wide level with differently located M-loci, which share a conserved Mdmd gene promoting male sex determination. Taking advantage of DNA long reads sequencing technologies, the authors produced massive sequence data and related analyses showing that all M-loci also have different molecular architectures and confirmed different cytogenetic localization in situ. By comparing the various genomic data, the authors reconstructed the potential evolutionary trajectory of the various M-loci. They concluded that they arose from unique independent translocations of large DNA fragments containing Mdmd from an ancient Y chromosome, leading to diverse M-loci genomic architectures. For example, the authors found convincing indications that MV was likely produced by transposase activity, while MIII was produced by double-strand breakage and homologous repair, typical of tandem duplications.

This work contributed to deepening our fundamental knowledge of the evolution of sex-determining primary signals at the genetic and chromosomal levels.

>We are happy that the reviewer acknowledges the significance of our study.

Minor suggestions follow.

Abstract

In the housefly, male-determining M-loci can reside on the Y chromosome 28 (MY), but also on the X chromosome (MX) and autosomes II (MII), III (MIII), V (MV). We investigated differently located M-loci, all containing the primary male-determining gene *Musca domestica* male determiner (Mdmd).

I suggest mentioning the *Musca Mdmd* master gene from the very start of the abstract and specifying if all Ms are Mdmd orthologues.

38 containing the same primary signal, independently diverged into regions of various complexity, leading to distinctly different male-determining loci.

I suggest to mention that the same primary signal is Mdmd.

>We have revised the abstract to include Mdmd mentioning.

A specific prediction of the canonical sex

50 chromosome evolution model is that a sex-determining region will undergo progressive recombination suppression to preserve its function³.

Specify here briefly why the sex-determining region requires progressive recombination suppression to preserve its function.

>We added a brief explanation in lines 55-56. "Suppressed recombination is predicted to prevent gene flow between proto-sex chromosomes so that the sex-determining region can be sex-limited and thus effectively haploid."

Results

As there was only a female genome available²², we assembled male genomes

74 from Pacbio SMRT sequencing of the strains M3 (~116x total coverage) and M5¹

79 addition, we performed Illumina sequencing on strains M3 (males and females), M5 (males), and M2 (males, which carry MII).

Why have the authors sequenced M3 in addition to M5? The experimental design description could be more straightforward.

Why have the authors sequenced by Illumina also M2 in addition to M3 and M5?

>We rearranged the description of results (see lines 83-89 and 122-135), so it is easier to follow the logic behind our experimental design.

83 As a first estimate of sequence divergence between M loci we determined the coverage of Mdmd. Please specify the correlation between the coverage of Mdmd and sequence divergence. Needs to be clarified.

>We added one explanatory sentence in lines 99-100. "Such coverages essentially represent Mdmd copy numbers in the tested M-loci and, therefore, are indicative of differences in the sizes of M genomic loci."

88 aabys male, ~19.62; A3-male, ~19.74) were similar. Interestingly, one MY dataset, LPR-male, showed a higher average coverage (~34.78) than the other two MY datasets.

What is LPR-male?

>We provided detailed information regarding MY strains in lines 92-95. "We also used published Illumina reads of three MY strains of different geographical origin²¹, aabys (laboratory generated strain with MY), A3 (strain with MY that was derived from a collection in Marshall County, Alabama, USA in 1998), and LPR (strain with MY that was originally collected near Horseheads, New York, USA)."

92 To identify the cytogenetic localization of M loci

I suggest adding the type of Musca tissues and cells the authors have used here.

>We added such information in line 109. "...we performed fluorescence in situ hybridization (FISH) with an Mdmd-specific probe and karyogram obtained from the brain tissues of third instar larvae".

98 not pericentromeric. Using samples from multiple laboratories
Which laboratories?

>We meant laboratory strains. This typo is fixed in line 115.

100 chromosomes, regardless of strain origin (supplementary fig. S1), indicating a single evolutionary origin of each of these M-loci
I would substitute the term "indicating" with "suggesting."

>Revised. See line 117.

105 The results above indicated that the complexity of MV was lower than that of the other M-loci
Please specify the meaning of the term complexity. Do the authors mean simply the copy number?

>By complexity, we meant sequence structures. As a higher Mdmd copy number not only suggests the MIII locus is larger in size, but also contains more non-Mdmd sequences intervening Mdmd copies. We changed it to "...the results indicated that the genomic sizes of MIII and MV were the most distinct" instead of using "complexity" in line 122.

109 estimated coverage of Mdmd for the M5 genome. Only a single synonymous SNP, located in exon 2, was found between these two Mdmd copies
SNP in ORF? Aminoacidic substitution observed?

>As we specified, the nucleotide difference is synonymous, therefore, there is no amino acid change. We also revised "SNP" to "nucleotide substitution" in line 141, which we think is more appropriate.

118 The ~31 kb sequence, which exists only on the MV-contig and includes the two opposing Mdmd copies, can thus be considered as the complete MV locus. MV is integrated in a ~10 kb tandem repeat block shared between the MV-contig and non-MV-contigs.
What is the sequence similarity of the 10Kb tandem repeat block shared between the homologous V chromosomes? Approximate divergence time?

>Based on BLAST search, the percentage identity between the repeat block on the homologous autosome V is ~99%. However, we think it is tricky to estimate divergence time based on variation between repeat sequences because they are not functionally conserved and are known to freely accumulate mutations. We also could not determine which variants appeared after the insertion of MV and which variants already occurred before.

162 MIII are likely degenerated pseudo-copies. None of these genes have been reported to be involved in sex determination.
Significantly few known genes involved in sex determination (hence as primary signals) have been isolated. These are different in various groups of insects. Please specify and expand the sentence "None of these genes have been reported to be involved in sex determination". Did the authors expect differently? In what sense?

>Although Mdmd has been confirmed to be the master male-determining gene in the housefly, it is possible that there are other genes tightly linked to Mdmd in the M locus, which also contribute to different stages of the male development or function as sexually antagonistic factors. However, this is less likely to be the case because as shown by our results, other putative gene sequences in MIII seem to be pseudo-copies. Moreover, considering that MV does not contain them, yet still functions fine as a male determiner, it suggests those pseudo-gene sequences are not essential.

185 the aforementioned results that the LPR male MY genome contains more M_{dmd} copies than the other male MY genomes.

Please give more information on LPR male and MY male genomes (Strains? Individuals of different origin?).

>We added detailed information in lines 93-95. LPR (strain with MY that was originally collected near Horseheads, New York, USA). To be more clear with what we meant, we also revised the LPR male MY genome to MY-locus in LPR male in line 235.

189 origin of these individual loci. Additionally, differences observed among MY imply that the duplication events on the M loci happened gradually and at different in frequencies.

Please correct at different in frequencies. Gradually? Or in subsequent steps?

>We changed it to in sequential steps, or that there was a large ancestral MY-locus, which was degraded differently in various strains, in line 236.

216 examine whether the MASX can be separately detected from MX, we performed Mix probe-based FISH on samples from two Spanish strains that possess an M-locus on the X chromosome. As no separate signals are observed (supplementary fig. S1i, j), MX is either included in or closely located to MASX. Please define MX as soon as it is introduced in the text, not later. It is confusing.

>Revised. See line 114, where MX is mentioned for the first time. We also added text at the beginning of the Results section for better clarity. In the following text, genomic regions with a dominant male-determining locus are referred to as M-loci with a superscript indicating on which chromosome the locus is found, i.e. M^{III} is the M-locus on chromosome III. Non-italic letter M is used to describe housefly strains or genomic datasets. M_{dmd} is the male-determining gene within all of the M loci investigated. See lines 77-80.

Before describing these data, a brief recap of the various strains having the M in various locations could be helpful.

>We briefly included the information in lines 258-260. We further tested the Mix probe with samples from strains aabys (males with MY), M5 (males with MV), and two Spanish strains (SPA1 and SPA4 in which samples possess MX).

248 Different mechanisms likely underlie the formation of distinctive M architectures. MV seems to have been created by transposase activity as we found sequence signatures of TIRs and DRs and highly similar

I would avoid using the term created and substitute it with produced, generated, etc.

>Revised. See line 308. MV seems to have been produced by transposase activity as we found sequence signatures...

Version 1:

Reviewer comments:

Reviewer #1

(Remarks to the Author)

The availability of MY lineage genome sequences represents a significant improvement in the paper. However, it is regrettable that the comparison of MASX and MASY sequences could not be conducted due to the low resolution of the assembly, possibly due to insufficient sequence coverage. As mentioned in previous comments and reiterated below, I consider this crucial for understanding the evolution of M loci. Therefore, I urge efforts to locate the corresponding regions from the PacBio raw sequences of the MY lineage, rather than being fixated on assembled contigs.

I still find the authors' claim that "the MY locus is the ancestor of the M3 and M5 loci" to be weak. I believe there is still the possibility that the M5 locus could be the origin. Below are the reasons for my stance:

1) The presence of tandem repeats downstream of Mdm is claimed to exist only in the M5 and MY loci, not in the M3 locus. However, if a similar tandem repeat exists at the end of M3-contig-2, it could suggest that the distance between Mdm and tandem repeat in the M3 locus has widened due to the overlapping MAS sequences, implying that tandem repeats are present in all M loci.

2) As emphasized in the initial manuscript, functional SNP analysis of Mdm indicates, most parsimoniously, that the M5 locus is closest to the inferred ancestral sequence. The six SNPs unique to the MY locus contradict the authors' hypothesis.

3) If the M5 locus is ancestral, no Mdm pseudogene should be inserted into the MASX of the M5 lineage. Conversely, if the MY locus is ancestral, there could be remnants of Mdm pseudogenes in the MASX of the M5 lineage. If so, these pseudogene repeats should be detectable by FISH with Mdm probes, but no signal is observed on the X chromosome of the M5 lineage.

If clear counterarguments cannot be provided for these points, fair consideration of the potentialities should be made.

Reviewer #2

(Remarks to the Author)

Following my previous comments, I am satisfied with the authors' replies and modifications. I suggest accepting and publishing the MS.

Author Rebuttal letter:

We thank the reviewers for their valuable comments. Below we provide answers to the reviewers' comments. We hope that our manuscript can now be accepted for publication in Nature Communications.

Reviewer #1 (Remarks to the Author):

The availability of MY lineage genome sequences represents a significant improvement in the paper. However, it is regrettable that the comparison of MASX and MAS Y sequences could not be conducted due to the low resolution of the assembly, possibly due to insufficient sequence coverage. As mentioned in previous comments and reiterated below, I consider this crucial for understanding the evolution of M loci. Therefore, I urge efforts to locate the corresponding regions from the PacBio raw sequences of the MY lineage, rather than being fixated on assembled contigs.

> We thank the reviewer for recognition of our effort to improve the manuscript.

We have clarified the origin of the MAS sequence in the Results (lines 242-245, lines 255-263), which may address some of the reviewer's concerns. The results now describe how the MAS were identified using a Mix probe that was generated from PCR products that were specifically amplified from the genomic DNA of the MIII-locus. The PCR products contain Mdm fragments and intercalated non-Mdm sequences within truncated Mdm copies in MIII.

We have also performed an analysis of the PacBio raw sequences, as suggested by the reviewer, to further compare MASX and MAS Y sequences. We mapped PacBio reads of aabys-male (with MY) and M5-male (with MV) to the MY-contigs. We identified regions of the MY contigs to which M5-male reads did not align, which we refer to as MY-specific regions. Intervening MY-specific regions are regions that show high sequence coverages from M5 male reads, of which some are likely repetitive sequences. We infer the mapped M5 reads come from genomic regions that share sequence identity with MY regions which are potentially on the X chromosome. This supports the hypothesis that the complex structure of MY was formed via repeated duplication of the Mdm gene on the Y chromosome, within a genomic region containing sequences found on both the X and Y. Therefore, our analysis of PacBio reads and FISH experiments paint a consistent picture that the M locus on the Y chromosome contains Mdm sequences interleaved with sequences that are shared between the X and Y. This supports our hypothesis described in response to the next question from the reviewer. We added the mapping results in supplementary material (fig. s7) and described it in discussion lines 305-310.

I still find the authors' claim that "the MY locus is the ancestor of the M3 and M5 loci" to be weak. I believe there is still the possibility that the M5 locus could be the origin.

> We agree with the reviewer that our results cannot fully support the ancestry of MY. However, the analysis of the MAS sequence described above supports the hypothesis that MY is the ancestor of MIII. The integration of MAS sequence with Mdm copies on the Y chromosome can be most parsimoniously explained by expansion of the M locus via duplication within a region shared between the X and Y. When this locus was duplicated from the Y to the chromosome III, it carried

with it both Mdm and the interleaved MAS sequences. Yet as the reviewer correctly points out, we cannot determine if MV is ancestral or derived relative to MY. We now provided two possible evolutionary scenarios of M-loci (MV-first vs. MY-first) in the discussion. See line 318-339. We discuss why we favor one over the other and how one can further investigate the evolutionary history of M-loci in future studies. We thoroughly considered the reviewer's arguments for MV being the first. Below are our perspectives on those stances.

Below are the reasons for my stance:

1) The presence of tandem repeats downstream of Mdm is claimed to exist only in the M5 and MY loci, not in the M3 locus. However, if a similar tandem repeat exists at the end of M3-contig-2, it could suggest that the distance between Mdm and tandem repeat in the M3 locus has widened due to the overlapping MAS sequences, implying that tandem repeats are present in all M loci.

> We infer the reason why tandem repeats (at the end of M3-contig-2) are distantly located from the intact Mdm copy, is because of additional duplication that occurred in MII subsequent to its translocation from MY. We agree it does not indicate the ancestry of the M-loci, as it assumes expansion by duplication took place after the establishment of MIII. The complex structure of M could have either developed from the primary Mdm that originated from a duplication of Mdn on the Y, or a duplicated Mdm that originated from MV.

2) As emphasized in the initial manuscript, functional SNP analysis of Mdm indicates, most parsimoniously, that the M5 locus is closest to the inferred ancestral sequence. The six SNPs unique to the MY locus contradict the authors' hypothesis.

> We do not think that using similarity to the Mdm consensus sequence to determine whether MV is ancestral is appropriate. As we discussed in the manuscript, there is a possibility of gene conversion between palindrome arms in MV (lines 275-279). Gene conversion can cause concerted evolution, whereby new mutations are eliminated and the ancestral state of sequences will be restored. The evolutionary time indicated by the number of mutations is thus unreliable.

3) If the M5 locus is ancestral, no Mdm pseudogene should be inserted into the MASX of the M5 lineage. Conversely, if the MY locus is ancestral, there could be remnants of Mdm pseudogenes in the MASX of the M5 lineage. If so, these pseudogene repeats should be detectable by FISH with Mdm probes, but no signal is observed on the X chromosome of the M5 lineage.

> The absence of Mdm pseudo-copies on the sex chromosomes of the M5 strains can be explained by what the reviewer suggested. However, there is an alternative explanation. MV could be derived from a cut-and-paste of the MY DNA fragment containing only intact Mdm. Then, as the reviewer mentioned, the Y chromosome should still carry Mdm truncated copies that could be detected by Mdm probes. However, in that case, the Y has lost the male determining function and the entire chromosome may have been lost in the population during the evolutionary process. Resulting individuals will only carry two X chromosomes that do not contain Mdm pseudo-copies. In fact, in nature, populations with hemizygous MIII, M-free X chromosomes, and absence of the Y are quite commonly found. Those cases represent scenarios of Y-loss after an alternative chromosome gained M locus. Hence, the lack of Mdm pseudo-copies on the X chromosome in the M5 strain cannot sufficiently support the ancestry of MV.

If clear counterarguments cannot be provided for these points, fair consideration of the potentialities should be made.

> We added discussion on the evolutionary history of the M-loci, including the reviewer's suggestions, in lines 318-339.

Version 2:

Reviewer comments:

Reviewer #1

(Remarks to the Author)

We feel that the revised manuscript provides a fair discussion. I appreciate the authors' efforts.
